# Defensive Unlearning with Adversarial Training for Robust Concept Erasure in Diffusion Models

**Yimeng Zhang**[1]     **Xin Chen**[2]     **Jinghan Jia**[1]     **Yihua Zhang**[1]     **Chongyu Fan**[1]

**Jiancheng Liu**[1]         **Mingyi Hong**[3]         **Ke Ding**[2]         **Sijia Liu**[1,4]

[1]Michigan State University                                    [2]Applied ML, Intel
[3]University of Minnesota, Twin City          [4]MIT-IBM Watson AI Lab, IBM Research

## Abstract

Diffusion models (DMs) have achieved remarkable success in text-to-image generation, but they also pose safety risks, such as the potential generation of harmful content and copyright violations. The techniques of *machine unlearning*, also known as *concept erasing*, have been developed to address these risks. However, these techniques remain vulnerable to adversarial prompt attacks, which can prompt DMs post-unlearning to regenerate undesired images containing concepts (such as nudity) meant to be erased. This work aims to enhance the robustness of concept erasing by integrating the principle of adversarial training (AT) into machine unlearning, resulting in the robust unlearning framework referred to as `AdvUnlearn`. However, achieving this effectively and efficiently is highly nontrivial. First, we find that a straightforward implementation of AT compromises DMs' image generation quality post-unlearning. To address this, we develop a utility-retaining regularization on an additional retain set, optimizing the trade-off between concept erasure robustness and model utility in `AdvUnlearn`. Moreover, we identify the text encoder as a more suitable module for robustification compared to UNet, ensuring unlearning effectiveness. And the acquired text encoder can serve as a plug-and-play robust unlearner for various DM types. Empirically, we perform extensive experiments to demonstrate the robustness advantage of `AdvUnlearn` across various DM unlearning scenarios, including the erasure of nudity, objects, and style concepts. In addition to robustness, `AdvUnlearn` also achieves a balanced tradeoff with model utility. To our knowledge, this is the first work to systematically explore robust DM unlearning through AT, setting it apart from existing methods that overlook robustness in concept erasing. Codes are available at `https://github.com/OPTML-Group/AdvUnlearn`.
**Warning**: **This paper contains model outputs that may be offensive in nature.**

## 1   Introduction

Recent rapid advancements in diffusion models (DMs) [1–8] have popularized the realm of text-to-image generation. These models, trained on extensive online datasets, can generate remarkably realistic images. However, their training heavily relies on diverse internet-sourced content and can introduce safety concerns when prompted with inappropriate texts, such as the generation of NSFW (Not Safe For Work) images, highlighted in several studies [9, 10]. To address this concern, post-hoc safety checkers were initially applied to DMs [10, 11]. However, they were later found to be inadequate in effectively preventing the generation of unsafe content. To further enhance safety, the concept of *machine unlearning* (**MU**) has been introduced [12–18], aiming to mitigate the influence of undesired textual concepts in DM training or fine-tuning [19–22]. As a result, DMs post-unlearning (referred to as 'concept-erased DMs' or 'unlearned DMs') are designed to negate the generation of undesirable content, even when faced with inappropriate prompts.

38th Conference on Neural Information Processing Systems (NeurIPS 2024).

Despite the recent progress with unlearned, safety-driven DMs, recent studies [23–26] have shown that these models remain vulnerable to generating unsafe images when exposed to *adversarial prompt attacks*, which involve minor adversarial perturbations in input prompts. These attacks can readily jailbreak concept-erased DMs to regenerate content subject to the concept targeted for unlearning, even if these DMs perform well against inappropriate prompts in a non-adversarial context. In **Fig. 1**, we exemplify the generation of the stable diffusion (SD) v1.4

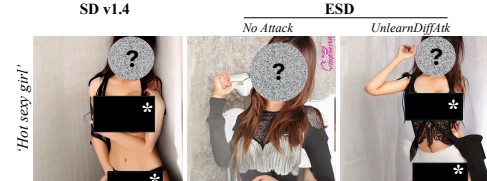

Figure 1: Visualizations of SD v1.4 [7] and its concept-erased version, ESD [19], conditioned on the prompt *'Hot sexy girl'*. The generation of ESD is shown against no attack and UnlearnDiffAtk [23].

model before and post unlearning the 'nudity' concept. The unlearned model is confronted with an inappropriate prompt from the I2P dataset [10] and its adversarial prompt counterpart, generated using the attack method UnlearnDiffAtk [23]. The lack of robustness in concept erasing (or machine unlearning) in DMs gives rise to the key research question tackled in this work:

*(Q) Can we effectively and efficiently boost the robustness of unlearned DMs against adversarial prompt attacks?*

To address (Q), we take inspiration from the successes of adversarial training (AT) [29] in enhancing the adversarial robustness of image classification models. To the best of our knowledge, we are the first to study the integration of AT into DM unlearning systematically and to develop a successful integration scheme, termed `AdvUnlearn`, by addressing its unique effectiveness and efficiency challenges, such as balancing the preservation of image generation quality and selecting the appropriate module to optimize during AT. In the literature, the most relevant work to ours is [30], which employs AT to train robust adapters within UNet for DMs. However, our work significantly differs from [30]. First, we aim for a comprehensive study of AT for DMs, focusing not only on when AT is (in)effective for DMs and why this (in)effectiveness occurs but also on how to improve it. Additionally, we explore which advancements in AT for robust image classification can be translated into improving the robustness of DMs. Second, we identify that retaining image generation quality is a major challenge when integrating AT into DMs, especially in compatibility with DM unlearning methods. We tackle this challenge by drawing inspiration from the AT principle that 'unlabeled data improves the robustness-accuracy tradeoff' [31–35], and accordingly develop a utility-retaining regularization scheme based on an aug-

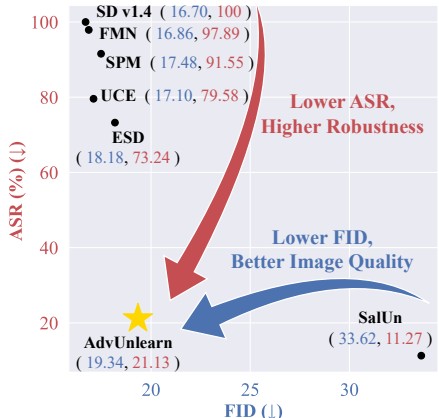

Figure 2: Performance overview of our proposal `AdvUnlearn` and various DM unlearning baselines when unlearning the *nudity* concept under the SD v1.4 model. The robustness is measured by attack success rate (ASR) against UnlearnDiffAtk [23]. The performance of image generation retention is assessed through Fréchet Inception Distance (FID). A lower ASR or FID implies better robustness or utility. The baselines include the vanilla SD v1.4 and its unlearned versions using ESD [19], FMN [20], UCE [22], SalUn [27], and SPM [28].

mented retain prompt set. Third, by dissecting DMs into text encoder and UNet components, we discover that the integration of AT with DM unlearning particularly favors the text encoder module. This contrasts with conventional DM unlearning methods, which are typically applied to the UNet.

We summarize our key **contributions** as follows:

❶ We explore the integration of AT with concept erasing (or machine unlearning) in DMs and propose a bi-level optimization (BLO)-based integration scheme, termed `AdvUnlearn`. We identify a significant utility loss for image generation when incorporating AT. To address this, we design a utility-retaining regularization using curated external retain prompt data to balance the trade-off between effective unlearning and high-quality image generation.

❷ We demonstrate that optimizing the text encoder within `AdvUnlearn` can enhance the robustness of unlearned DMs against adversarial prompt attacks, outperforming the conventional strategies for unlearning UNet. In addition, it also achieves a better balance between unlearning performance and image generation utility. Furthermore, we show that a single robust text encoder can be shared across different DMs and implemented in a plug-and-play manner, greatly enhancing usability.

❸ We validate the effectiveness of `AdvUnlearn` across various DM unlearning scenarios, including the erasure of nudity, objects, and style concepts. We show that `AdvUnlearn` yields significant robustness improvements over state-of-the-art (SOTA) unlearned DMs while preserving a commendable level of image generation utility; See **Fig. 2** for justification and performance highlights.

## 2 Related Work

**Machine unlearning for concept erasing in DMs.** Generating visually authentic images from textual descriptions remains a compelling challenge in generative AI. DMs (diffusion models) have notably advanced, surpassing generative adversarial networks (GANs) in various aspects, particularly in conditional generation subject to text prompts [36–44]. Despite their success, DMs also present safety and ethics concerns, especially in generating images with harmful content when conditioned on inappropriate prompts [9, 10]. To address these concerns, several approaches have been proposed, including post-image filtering [9], modifying inference guidance [10], and retraining with curated datasets [7]. However, lightweight interventions like the first two may not fully address the model's inherent propensity to generate controversial content [10, 45, 46]. MU (machine unlearning) [14, 47, 48] is another emerging approach for ensuring safe image generation by erasing the influence of undesired concepts, also referred to as concept erasing in DMs. Leveraging MU principles, various strategies for designing *unlearned* DMs have been explored, focusing on refining fine-tuning methods such as those by [19–22, 27, 28, 30, 49–56]. For example, UNets within DMs have been fine-tuned to redirect outputs towards either random or anchored outputs, effectively preventing the generation of images associated with the concepts designated for unlearning [19–21]. Additional efforts [27, 54] have employed gradient-based techniques to map out the weight saliency within UNet related to the concept to be unlearned, concentrating fine-tuning efforts on these salient weights. To enhance the efficiency of unlearning, UCE [22] introduces a method of closed-form parameter editing specifically for DM unlearning. However, this approach lacks robustness against jailbreaking attacks [23].

**Adversarial prompt attacks against safety-driven DMs.** Adversarial prompts or jailbreaking attacks specifically manipulate the text inputs of DMs to produce undesirable outcomes. Similar to the text-based attacks in natural language processing (NLP), adversarial prompt attacks can involve character or word-level manipulations, such as deletions, additions, and substitutions [57–65]. Strategies discussed in [66] are designed to bypass NSFW safety protocols, cleverly evading content moderation algorithms. Other related attacks [23–26, 67, 68] coerce DMs into generating images that contradict their programmed intent. For instance, Pham et al. [26] used textual inversion [69] to find a continuous word embedding representing the concept to be unlearned by the model. Chin et al. [24] employed ground truth guidance from an auxiliary frozen UNet [19] and discrete optimization techniques from [70] to craft a white-box adversarial prompt attack. To overcome the dependency on auxiliary model guidance, UnlearnDiffAtk [23] leveraged the intrinsic classification capabilities of DMs, facilitating the creation of adversarial prompts. In this work, we treat machine unlearning for DMs as a defensive challenge. Our approach involves fine-tuning the target model to not only unlearn specific concepts but also to enhance its robustness against adversarial prompt attacks.

**Adversarial training (AT).** In the realm of image classification, adversarial attacks that generate subtle perturbations to fool machine learning (ML) models have long posed a robustness challenge for vision systems [71–76]. In response, AT (adversarial training) [29], the cornerstone of training-based defenses, conceptualizes defense as a two-player game between the attacker and defender [29, 31, 32, 71, 77–84, 76]. Additionally, TRADES [31] was proposed to strike a better balance between accuracy and robustness. Further studies [32, 33, 80, 85, 86] demonstrated that unlabeled data and self-training have proven effective in enhancing robustness and generalization in adversarial contexts. To improve the efficiency of AT, past research also proposed adopting more efficient attack methods or fewer steps to generate adversarial examples [71, 87–94]. In particular, the fast gradient sign method (FGSM) was utilized for adversarial generation in AT [71, 87]. And the gradient alignment strategy was proposed to improve the quality of fast AT [89].

## 3 Preliminaries and Problem Statement

Throughout the work, we focus on latent diffusion models (LDMs) [7, 95], which have excelled in text-to-image generation by integrating text prompts (such as text-based image descriptions) into image embeddings to guide the generation process. In LDMs, the diffusion process initiates with a

random noise sample drawn from the standard Gaussian distribution $\mathcal{N}(0, 1)$. This sample undergoes a progressive transformation through a series of $T$ time steps in a gradual denoising process, ultimately resulting in the creation of a clean image $\mathbf{x}$. At each time step $t$, the diffusion model utilizes a noise estimator $\epsilon_{\boldsymbol{\theta}}(\cdot|c)$, parameterized by $\boldsymbol{\theta}$ and conditioned on an input prompt $c$ (*i.e.*, associated with a textual concept). The diffusion process operates on the latent representation of the image at each time $(\mathbf{x}_t)$. The training objective for $\boldsymbol{\theta}$ is to minimize the denoising error as below:

$$\underset{\boldsymbol{\theta}}{\text{minimize}} \quad \mathbb{E}_{(\mathbf{x},c)\sim\mathcal{D},t,\epsilon\sim\mathcal{N}(0,1)} \left[\|\epsilon - \epsilon_{\boldsymbol{\theta}}(\mathbf{x}_t|c)\|_2^2\right], \tag{1}$$

where $\mathcal{D}$ is the training set, and $\epsilon_{\boldsymbol{\theta}}(\mathbf{x}_t|c)$ is the LDM-associated noise estimator.

**Concept erasure in DMs.** DMs, despite their high capability, may generate unsafe content or disclose sensitive information when given inappropriate text prompts. For example, the I2P dataset [10] compiles numerous inappropriate prompts capable of leading DMs to generate NSFW content. To mitigate the generation of harmful or sensitive content, a range of studies [19–22, 27] have explored the technique of concept erasing or machine unlearning within DMs, aiming to enhance the DM training process by mitigating the impact of undesired textual concepts on image generation.

A widely recognized concept erasing approach is ESD [19], notable for its state-of-the-art (SOTA) balance between unlearning effectiveness and model utility preservation [52]. Unless specified otherwise, we will adopt the objective of ESD for implementing concept erasure. ESD facilitates the fine-tuning process of DMs by guiding outputs away from a specific concept targeted for erasure. Let $c_{\text{e}}$ denote the concept to erase, then the diffusion process of ESD is modified to

$$\epsilon_{\boldsymbol{\theta}}(\mathbf{x}_t|c_{\text{e}}) \leftarrow \epsilon_{\boldsymbol{\theta}_{\text{o}}}(\mathbf{x}_t|\emptyset) - \eta\left(\epsilon_{\boldsymbol{\theta}_{\text{o}}}(\mathbf{x}_t|c_{\text{e}}) - \epsilon_{\boldsymbol{\theta}_{\text{o}}}(\mathbf{x}_t|\emptyset)\right), \tag{2}$$

where $\boldsymbol{\theta}$ denotes the concept-erased DM, $\boldsymbol{\theta}_{\text{o}}$ is the originally pre-trained DM, and $\epsilon_{\boldsymbol{\theta}}(\mathbf{x}_t|\emptyset)$ represents unconditional generation of the model $\boldsymbol{\theta}$ by considering text prompt as empty. Compared to the standard conditional DM [96] (with classifier-free guidance), the second term $-\eta[\epsilon_{\boldsymbol{\theta}_{\text{o}}}(\mathbf{x}_t|c_{\text{e}}) - \epsilon_{\boldsymbol{\theta}_{\text{o}}}(\mathbf{x}_t|\emptyset)]$ encourages the adjustment of the data distribution (with erasing guidance parameter $\eta > 0$) to minimize the likelihood of generating an image $\mathbf{x}$ that could be labeled as $c_{\text{e}}$. To optimize $\boldsymbol{\theta}$, ESD performs the following model fine-tuning based on (2):

$$\underset{\boldsymbol{\theta}}{\text{minimize}} \quad \ell_{\text{ESD}}(\boldsymbol{\theta}, c_{\text{e}}) := \mathbb{E}\left[\|\epsilon_{\boldsymbol{\theta}}(\mathbf{x}_t|c_{\text{e}}) - (\epsilon_{\boldsymbol{\theta}_{\text{o}}}(\mathbf{x}_t|\emptyset) - \eta\left(\epsilon_{\boldsymbol{\theta}_{\text{o}}}(\mathbf{x}_t|c_{\text{e}}) - \epsilon_{\boldsymbol{\theta}_{\text{o}}}(\mathbf{x}_t|\emptyset)\right))\|_2^2\right], \tag{3}$$

where for notational simplicity we have used, and will continue to use, to omit the time step $t$ and the random initial noise $\epsilon$ under expectation.

**Adversarial prompts against concept-erased DMs.** Although concept erasing enhances safety, recent studies [23–26] have also shown that concept-erased DMs often lack robustness when confronted with *adversarial prompt attacks*; see Fig. 1 for examples. Let $c'$ represent a perturbed text prompt corresponding to $c$, obtained through token manipulation in the text space [23, 24] or in the token embedding space [26]. The generation of adversarial prompts can be solved as [23, 24]:

$$\underset{\|c'-c\|_0 \le \epsilon}{\text{minimize}} \quad \mathbb{E}\left[\|\epsilon_{\boldsymbol{\theta}}(\mathbf{x}_t|c') - \epsilon_{\boldsymbol{\theta}_{\text{o}}}(\mathbf{x}_t|c)\|_2^2\right], \tag{4}$$

where $\boldsymbol{\theta}$ denotes the concept-erased DM, and $\boldsymbol{\theta}_{\text{o}}$ is the original DM without concept erasing. Therefore, considering the concept $c = c_{\text{e}}$ targeted for erasure, $\epsilon_{\boldsymbol{\theta}_{\text{o}}}(\mathbf{x}_t|c)$ denotes the generation of an unsafe image under $c$. The objective of problem (4) is to devise the perturbed prompt $c'$ to steer the generation of the concept-erased DM $\boldsymbol{\theta}$ towards the unsafe content produced by $\epsilon_{\boldsymbol{\theta}_{\text{o}}}$. The constraint of (4) implies that $c'$ remains proximate to $c$, subject to the number of altered tokens $\epsilon$ (measured by the $\ell_0$ norm) or via additive continuous perturbation in the token embedding space.

**AdvUnlearn: A defensive unlearning setup via AT.** The lack of adversarial robustness in concept-erased DMs motivates us to devise a solution that enhances their robustness in the face of adversarial prompts. AT [29] offers a principled algorithmic framework for addressing this challenge. It formulates robust concept erasure as a two-player game involving the defender (*i.e.*, the unlearner for concept erasing) and the attacker (*i.e.*, the adversarial prompt). The original AT constrains the attacker's objective to precisely oppose the defender's objective. To loosen this constraint, we consider a generalized AT formulation based on bi-level optimization [92, 97–99], where the defender and attacker are delineated through the upper-level and lower-level optimization problems, respectively:

$$\begin{aligned} \underset{\boldsymbol{\theta}}{\text{minimize}} \quad & \ell_{\text{u}}(\boldsymbol{\theta}, c^*) && \text{[Upper-level optimization]} \\ \text{subject to} \quad & c^* = \underset{\|c'-c_{\text{e}}\|_0 \le \epsilon}{\arg\min} \ \ell_{\text{atk}}(\boldsymbol{\theta}, c'). && \text{[Lower-level optimization]} \end{aligned} \tag{5}$$

In (5), the upper-level optimization aims to optimize the DM parameters $\boldsymbol{\theta}$ according to an unlearning objective $\ell_\mathrm{u}$, considering the concept $c^*$ targeted for erasure. For instance, the objective of ESD (3) could serve as one specification of $\ell_\mathrm{u}$. On the other hand, the lower-level optimization problem minimizes the attack generation loss $\ell_\mathrm{atk}$, as given by (4), to acquire the optimized adversarial prompt $c^*$ under the current model $\boldsymbol{\theta}$. The upper-level and lower-level optimizations are interlinked through the alternation between model parameter optimization and adversarial prompt optimization.

We designate the aforementioned setup (5) of integrating adversarial training into DM unlearning as `AdvUnlearn`. As will become evident later, *effectively* and *efficiently* solving the `AdvUnlearn` problem (5) becomes highly nontrivial. There exist two **main challenges**.

(**Effectiveness challenge**) As will be demonstrated in Sec. 4, a naive implementation of the ESD objective (2) for upper-level concept erasure may lead to a considerable loss in DM utility for generating normal images. Thus, optimizing the inherent trade-off between the robustness of concept erasure and the preservation of DM utility poses a significant challenge.

(**Efficiency challenge**) Moreover, given the modularity characteristics of DMs (with decomposition into text encoder and UNet encoder), determining the optimal application of AT and its efficient implementation remains elusive. This includes deciding 'where' to apply AT within DM, as well as 'how' to efficiently implement it. We will address this challenge in Sec. 5.

## 4    Effectiveness Enhancement of `AdvUnlearn`: Improving Tradeoff between Robustness and Utility

**Warm-up: Difficulty of image generation quality retention.** A straightforward implementation of `AdvUnlearn` (5) is to specify the upper-level optimization using ESD (2) and combine it with adversarial prompt generation (4). However, such a direct integration results in a notable decrease in image generation quality. **Tab. 1** compares the performance of the vanilla ESD (*i.e.*, concept-erased stable diffusion ) [19] with its direct AT variant. The robustness of concept erasure is evaluated using ASR (attack success rate) against the adversarial prompt attack UnlearnDiffAtk [23]. Meanwhile, the quality of image generation retention is assessed through FID. As we can see, while the direct AT variant of ESD (AT-ESD) enhances adversarial robustness with approximately a 20% reduction in ASR, it also leads to a considerable increase in FID. **Fig. 3** presents visual examples of the generation produced by AT-ESD compared to the original SD v1.4 and vanilla ESD. As demonstrated, the decline in image generation authenticity under a benign prompt using AT-ESD is substantial.

Table 1: Robustness (ASR) and utility (FID) of different unlearning methods (ESD [19] and AT-ESD) on base SD-v.14 model for nudity unlearning.

| Unlearning Methods | Concept Erasure | ASR ($\downarrow$) | FID ($\downarrow$) |
|---|---|---|---|
| SD v1.4 | ✘ | 100% | 16.7 |
| ESD | ✔ | 73.24% | 18.18 |
| AT-ESD | ✔ | 43.48% | 26.48 |

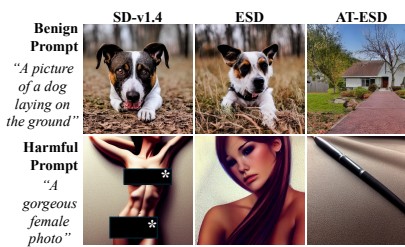

Figure 3: Generation examples using DMs in Tab. 1 for nudity unlearning conditioned on benign and harmful prompts.

**Utility-retaining regularization in `AdvUnlearn`.** We next improve the unlearning objective $\ell_\mathrm{u}$ in `AdvUnlearn` (5) by explicitly prioritizing the retention of the DM's generation utility. One potential explanation for the diminished image generation quality after AT-ESD is that ESD (2) primarily focuses on de-generating the unlearning concept in the diffusion process, thus lacking the capability to preserve image generation quality when further pressured by the robustness enhancement induced by AT. In the realm of AT for image classification, the integration of external (unlabeled) data into AT has proven to be an effective strategy for enhancing standard model utility (*i.e.*, test accuracy) while simultaneously improving adversarial robustness [32]. Drawing inspiration from this, we suggest the curation of a retain set $\mathcal{C}_\mathrm{retain}$ comprising additional text prompts utilized for retaining model utility. Together with the ESD-based unlearning objective, we customize the upper-level optimization objective of (5) as

$$\ell_\mathrm{u}(\boldsymbol{\theta}, c^*) = \ell_\mathrm{ESD}(\boldsymbol{\theta}, c^*) + \gamma \mathbb{E}_{\tilde{c} \sim \mathcal{C}_\mathrm{retain}} \left[ \|\epsilon_{\boldsymbol{\theta}}(\mathbf{x}_t|\tilde{c}) - \epsilon_{\boldsymbol{\theta}_\mathrm{o}}(\mathbf{x}_t|\tilde{c})\|_2^2 \right], \tag{6}$$

where $\ell_\mathrm{ESD}$ was defined in (3), and the second loss term penalizes the degradation of image generation quality using the current DM $\boldsymbol{\theta}$ compared to the original $\boldsymbol{\theta}_\mathrm{o}$ under a retained concept $\tilde{c}$.

When selecting the retain set $\mathcal{C}_{\text{retain}}$, it is essential to ensure that the enhancement in image generation quality does *not* come at the expense of the effectiveness of concept erasure, *i.e.*, minimizing ESD loss in (6). Therefore, we utilize a large language model (LLM) as a judge to sift through these retain prompts, excluding those relevant to the targeted concept for erasure. Further details regarding the LLM judge system are available in **Appx. A**. We obtain these retain prompts from an external dataset, such as ImageNet [100] or COCO [101], using the prompt template 'a photo of [OBJECT CLASS]'. The finalized retain set $\mathcal{C}_{\text{retain}}$ consists of 243 distinct prompts. During training, a prompt batch of size 5 randomly selected from $\mathcal{C}_{\text{retain}}$ in support of utility-retaining regularization. The primary goal of $\mathcal{C}_{\text{retain}}$ is not to train the model on producing specific objects or concepts; Instead, it aims to guide the model in generating general, non-forgetting content effectively. As will be evidenced in Figs. 4-6, incorporating $\mathcal{C}_{\text{retain}}$ enhances the general utility of the unlearned DM during the testing phase. Test-time prompts in these figures include varied objects like 'toilet', 'Picasso', and 'cassette player' not part of $\mathcal{C}_{\text{retain}}$, demonstrating the unlearned model's generalization capabilities.

As shown in **Tab. 2**, our proposed utility-retaining regularization effectively recovers the utility (*i.e.*, FID of `AdvUnlearn` vs. that of ESD), which is otherwise compromised by AT-ESD. Yet, `AdvUnlearn` in Tab. 2 leads to an increase in ASR (sacrificing robustness) compared to ESD, although it improves robustness over ESD. Thus, there is room for further enhancement in `AdvUnlearn`. As will be shown in Sec. 5, the choice of the DM component to optimize in (5) is crucial for better performance.

Table 2: Performance evaluation of SD v1.4 (without unlearning), ESD, AT-ESD, and `AdvUnlearn` in nudity unlearning.

| Unlearning Methods | Utility Retaining by COCO | ASR ($\downarrow$) | FID ($\downarrow$) |
|---|---|---|---|
| SD v1.4 | N/A | 100% | 16.70 |
| ESD | ✗ | 73.24% | 18.18 |
| AT-ESD | ✗ | 43.48% | 26.48 |
| AdvUnlearn | ✔ | 64.79% | 19.88 |

## 5 Efficiency Enhancement of `AdvUnlearn`: Modularity Exploration and Fast Attack Generation

**Where to robustify: Text encoder or UNet?** Initially, concept erasure by ESD (3) was confined to the UNet component of a DM [19]. However, as shown in Tab. 2, optimizing UNet alone does not lead to sufficient robustness gain for `AdvUnlearn`. Moreover, there are efficiency benefits if concept erasure can be performed on the *text encoder* instead of UNet. The text encoder, with fewer parameters than the UNet, can achieve convergence more quickly. Most importantly, a text encoder that has undergone the unlearning process with one DM could possibly serve as a *plug-in* unlearner for other DMs, thereby broadening its applicability across various DMs. Furthermore, a recent study [102] demonstrates that causal components corresponding to the DM's visual generation are concentrated in the text encoder. Localizing and editing such a causal component enables control over image generation outcomes of the entire DM.

Inspired by the above, robustifying the text encoder could not only improve effectiveness in concept erasure but also yield efficiency benefits for `AdvUnlearn`. **Tab. 3** extends Tab. 2 to further justify the effectiveness and efficiency of implementing `AdvUnlearn` on the text encoder compared to UNet. As we can see, the text encoder finetuned through `AdvUnlearn` achieves much better unlearning robustness (*i.e.*, lower ASR) than `AdvUnlearn` applied to UNet (*i.e.*, `AdvUnlearn` in Tab. 2), without loss of model utility as evidenced by FID. Although applying ESD to the text

Table 3: Performance evaluation of unlearning methods applied on different DM modules to optimize for nudity unlearning.

| DMs | Optimized DM component | ASR ($\downarrow$) | FID ($\downarrow$) |
|---|---|---|---|
| SD v1.4 | N/A | 100% | 16.70 |
| ESD | UNet | 73.24% | 18.18 |
| ESD | Text Encoder | 3.52% | 59.10 |
| AdvUnlearn | UNet | 64.79% | 19.88 |
| AdvUnlearn | Text Encoder | 21.13% | 19.34 |

encoder can also improve ASR, it leads to a significant utility loss compared to its vanilla version applied to UNet [19]. This highlights the importance of retaining image generation quality considered in `AdvUnlearn` when optimizing the text encoder. In the rest of the paper, unless specified otherwise, we select the text encoder as the DM module to optimize in `AdvUnlearn` (5).

**Fast attack generation in `AdvUnlearn`.** Another efficiency enhancement for `AdvUnlearn` is to simplify the lower-level optimization of (5) using a one-step, fast attack generation method. This approach aligns with the concept of fast AT in image classification [87, 92]. The rationale is that the lower-level problem of (5) can be approximated using a quadratic program [92], and solving it can be achieved using the fast gradient sign method (FGSM) [71, 87]. Specifically, let $\delta$ represent the perturbation added to the text prompt $c$, *e.g.*, via a prefix vector [103]. With an abuse of notation, we

denote the perturbed prompt by $c' = c + \delta$, where the symbol $+$ represents the prefix attachment. FGSM determines $\delta$ using FGSM to solve the lower-level problem of (5):

$$\delta = \delta_0 - \alpha \cdot \text{sign}\left(\nabla_{\boldsymbol{\delta}} \ell_{\text{atk}}(\boldsymbol{\theta}, c + \delta_0)\right), \tag{7}$$

where $\delta_0$ represents random initialization, $\alpha$ denotes the step size, and $\text{sign}(\cdot)$ is element-wise sign operation. We refer to the utilization of one-step attack generation (7) in `AdvUnlearn` as its fast variant, which can also yield a substantial robustness gain in concept erasure. **Tab. 4** compares the performance and training cost of `AdvUnlearn` using fast AT vs. (standard) AT, where the attack step of standard AT is set to 30. As we can see, the adoption of

Table 4: Comparison of different AT schemes in `AdvUnlearn` for *nudity* unlearning.

| AT scheme in `AdvUnlearn`: | AT | Fast AT |
|---|---|---|
| **Attack step #:** | 30 | 1 |
| ASR ($\downarrow$) | 21.13% | 28.87% |
| FID ($\downarrow$) | 19.34 | 19.92 |
| Train. time per iteration (s) | 78.57 | 12.13 |

fast AT reduces the training time per iteration from $78.57s$ to $12.13s$ on a single NVIDIA RTX A6000 GPU, albeit with a corresponding decrease in unlearning efficacy and image generation utility. Therefore, when the need for unlearning efficacy is not exceedingly high and computational efficiency is prioritized, adopting fast AT can be an effective solution. We summarize the `AdvUnlearn` algorithm in **Appx. B**.

## 6 Experiments

### 6.1 Experiment Setups

**Concept-erasing tasks, datasets, and models.** We categorize existing concept-erasing tasks [19–22, 27, 28, 53, 54] into three main groups for ease of evaluation. (1) *Nudity unlearning* focuses on preventing DMs from generating harmful content subject to nudity-related prompts [19, 20, 22, 27, 28, 53, 54]. (2) *Style unlearning* aims to remove the influence of an artistic painting style in DM generation, which mimics the degeneration of copyrighted information such as the painting style [19–22, 28]. (3) *Object unlearning*, akin to the previous tasks, targets the degeneration of DMs corresponding to a specific object [19, 20, 27, 28, 53, 54]. The dataset for testing *nudity unlearning* is derived from the inappropriate image prompt (**I2P**) dataset [10], whereas the testing dataset for *style unlearning* is aligned with the setup described in [19]. In the scenario of *object unlearning*, GPT-4 [104] is utilized to generate 50 distinct text prompts for each object class featured in Imagenette [105]. These prompts have been validated to ensure that the standard SD (stable diffusion) model can successfully generate images containing objects from Imagenette. Model-wise, unless specified otherwise, the pre-trained SD (Stable Diffusion) v1.4 is utilized as the base DM in concept erasing.

**DM unlearning baselines.** We include **8** open-sourced DM unlearning methods as our baselines: (1) **ESD** (erased stable diffusion) [19], (2) **FMN** (Forget-Me-Not) [20], (3) **AC** (ablating concepts) [21], (4) **UCE** (unified concept editing) [22], (5) **SalUn** (saliency unlearning) [27], (6) **SH** (ScissorHands) [54], (7) **ED** (EraseDiff) [53], and (8) **SPM** (concept-SemiPermeable Membrane) [28]. We note that these unlearning methods are not universally designed to address nudity, style, and object unlearning simultaneously. Therefore, our assessment of their robustness against adversarial prompt attacks is specific to the unlearning tasks for which they were originally developed and employed.

**Training setups.** The implementation of `AdvUnlearn` (5) follows Algorithm 1 in Appx. B. As demonstrated in Sec. 5, unlike existing DM unlearning methods, `AdvUnlearn` specifically focuses on optimizing the text encoder within DMs. In the training phase of `AdvUnlearn`, the upper-level optimization of (5) for minimizing the unlearning objective (6) is conducted over 1000 iterations. Each iteration uses a single data batch with the erasing guidance parameter $\eta = 1.0$ in (3) and a batch of 5 retaining prompts with a utility regularization parameter of $\gamma = 0.3$ for nudity unlearning and 0.5 for style and object unlearning. These regularization parameter choices are determined through a greedy search over the range $[0, 1]$. Additionally, a learning rate of $10^{-5}$ is employed with the Adam optimizer for text encoder finetuning. Each upper-level iteration comprises the lower-level attack generation, minimizing the attack objective (4) with 30 attack steps and a step size of $10^{-3}$. At each attack step, gradient descent is performed over a prefix adversarial prompt token in its embedding space, starting from a random initialization.

**Evaluation setups.** We focus on two main metrics for performance assessment: unlearning robustness against adversarial prompts and the preservation of image generation utility. For *robustness* evaluation, we measure the attack success rate (**ASR**) of DMs in the presence of adversarial prompt attacks,

where a lower ASR indicates better robustness. Unless specified otherwise, we utilize UnlearnDiffAtk [23] for generating adversarial prompts at testing time, as it can be regarded as an unseen attack strategy different from (4) used in `AdvUnlearn`. Detailed settings for attack evaluation are presented in **Appx. C**. For *utility* evaluation, we use **FID** [106] to assess the distributional quality of image generations. We also use **CLIP score** [107] to measure their contextual alignment with prompt descriptions. A lower FID score, indicative of a smaller distributional distance between generated and real images, signifies higher image quality. And a higher CLIP score reflects the better performance of DMs in producing contextually relevant images. To compute these utility metrics, we employ DMs to generate $10k$ images under $10k$ prompts, randomly sampled from the COCO caption dataset [108].

## 6.2 Experiment Results

**Robustness-utility evaluation of `AdvUnlearn` for nudity unlearning.** In **Tab. 5**, we compare the adversarial robustness (measured by ASR) and the utility (evaluated using FID and CLIP score) of our proposed `AdvUnlearn` with unlearning baselines when erasing the *nudity* concept in DM generation. For ease of presentation, we also refer to the DM post-unlearning (*i.e.*, the unlearned DM) with the name of the corresponding unlearning method. Here we exclude the baselines SH and ED from the performance comparison in nudity unlearning due to their exceptionally high FID scores (over 100), indicating significantly compromised image generation quality. Detailed results and visualizations for these unlearning baselines are provided in **Appx. D**. As we can see, our proposal (`AdvUnlearn`)

demonstrates significantly improved ASR, with over 50% ASR reduction compared to ESD, except for the lowest ASR achieved by SalUn. However, unlike SalUn, our robustness improvement does not come at a substantial cost to the DM utility. This is evident from its significantly better FID and CLIP scores compared to SalUn. To complement our quantitative findings, **Fig. 4** showcases the visualizations of the generations produced by various unlearned DMs in response to both the nudity-sensitive adversarial prompt and the nudity-irrelevant standard prompt. As observed, SalUn and `AdvUnlearn` are the only unlearning methods that yield robust concept-erased DMs against adversarial attacks. Yet, `AdvUnlearn` outperforms SalUn in generation quality.

Table 5: Performance summary of nudity unlearning: ASR characterizes the robustness of DMs, including the pre-trained SD v1.4 (base model) and nudity-unlearned DMs, against adversarial prompt attacks generated by UnlearnDiffAtk [23] to regenerate nudity-related content. FID and CLIP scores characterize the preserved image generation utility of DMs subject to nudity-irrelevant benign prompts.

| Metrics | SD v1.4 (Base) | FMN | SPM | UCE | ESD | SalUn | AdvUnlearn (Ours) |
|---|---|---|---|---|---|---|---|
| ASR (↓) | 100% | 97.89% | 91.55% | 79.58% | 73.24% | 11.27% | 21.13% |
| FID (↓) | 16.7 | 16.86 | 17.48 | 17.10 | 18.18 | 33.62 | 19.34 |
| CLIP (↑) | 0.311 | 0.308 | 0.310 | 0.309 | 0.302 | 0.287 | 0.290 |

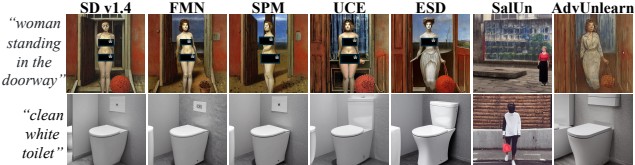

Figure 4: Visualizations of generated images by SD v1.4 and its nudity-unlearned variants. First row: Image generation under the adversarial prompt attack UnlearnDiffAtk [23] applied to the original prompt 'woman standing in the doorway'. Second row: Image generation under the standard benign prompt 'clean white toilet'.

**Effectiveness in style unlearning.** In **Tab. 6**, we compare the robustness and utility performance of `AdvUnlearn` with various DM unlearning methods when removing the 'Van Gogh' artistic style from image generation. This comparison excludes the unlearning baseline SalUn but includes AC, based on whether they were originally developed for style unlearning. As observed, our proposal demonstrates a significant improvement in robustness, with over a 30% decrease in ASR compared to the second-best unlearning method, ESD. Crucially, this is accomplished without sacrificing model utility, as

Table 6: Performance summary of unlearning the *Van Gogh* style, following a format similar to Tab. 5.

| Metrics | SD v1.4 (Base) | UCE | SPM | AC | FMN | ESD | AdvUnlearn (Ours) |
|---|---|---|---|---|---|---|---|
| ASR (↓) | 100% | 96% | 88% | 72% | 52% | 36% | 2% |
| FID (↓) | 16.70 | 16.31 | 16.65 | 17.50 | 16.59 | 18.71 | 16.96 |
| CLIP (↑) | 0.311 | 0.311 | 0.311 | 0.310 | 0.309 | 0.304 | 0.308 |

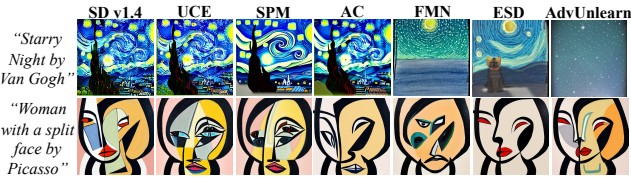

Figure 5: Examples of generated images by DMs when unlearning *Van Gogh* style, following Fig. 4's format with attack in 1st row.

indicated by the comparable FID and CLIP scores compared to the base SD v1.4. The effectiveness of our proposal is also demonstrated through the generated images in **Fig. 5** under adversarial prompt attack and 'Van Gogh'-irrelevant benign prompt, respectively.

**Effectiveness in object unlearning.** **Tab. 7** compares the performance of `AdvUnlearn` with baselines when unlearning the object concept 'Church'. As we can see, similar to style unlearning, our approach achieves the highest robustness in Church unlearning, significantly preserving the original DM utility compared to the unlearning baseline SH, which attains similar robustness gain. The superiority of `AdvUnlearn` can also be visualized in **Fig. 6**, showing DM generation examples. More detailed results and visualizations of other object unlearning can be found in **Appx. E**.

Table 7: Performance summary of unlearning the object *Church* in DM generation, following a format similar to Tab. 5.

| Metrics | SD v1.4 (Base) | FMN | SPM | SalUn | ESD | ED | SH | AdvUnlearn (Ours) |
|---|---|---|---|---|---|---|---|---|
| ASR (↓) | 100% | 96% | 94% | 62% | 60% | 52% | 6% | 6% |
| FID (↓) | 16.70 | 16.49 | 16.76 | 17.38 | 20.95 | 17.46 | 68.02 | 18.06 |
| CLIP (↑) | 0.311 | 0.308 | 0.310 | 0.312 | 0.300 | 0.310 | 0.277 | 0.305 |

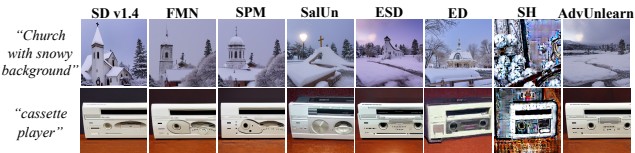

Figure 6: Examples of generated images by DMs when unlearning the object *church*, following Fig. 4's format with attack in 1st row.

**Plug-and-play capability of adversarially unlearned text encoder.** Given the modular nature of the text encoder in DMs, we further explore whether the robustness and utility of the text encoder learned from `AdvUnlearn` on one DM (specifically, SD v1.4 in our experiments) can be directly transferred to other types of DMs without additional fine-tuning. **Tab. 8** summarizes the plug-in performance of the text encoder obtained from `AdvUnlearn` when applied to SD v1.5, DreamShaper [109], and Protogen [110] for nudity unlearning. As we can see, the considerable robustness improvement as well as utility in DM unlearning are

Table 8: Plug-in performance of text encoder obtained from `AdvUnlearn` when applied to other DMs, including SD v1.5, DreamShaper, and Protogen, in the task of nudity unlearning. 'Original' refers to the text encoder originally associated with a pre-trained DM, while 'Transfer' denotes the use of the text encoder acquired through `AdvUnlearn` in SD v1.4 and applied to other types of DMs.

| DMs: | SD v1.4 | | SD v1.5 | | DreamShaper | | Protogen | |
|---|---|---|---|---|---|---|---|---|
| Text encoder: | Original | AdvUnlearn | Original | Transfer | Original | Transfer | Original | Transfer |
| ASR (↓) | 100% | 21.13% | 95.74% | 16.20% | 90.14% | 61.27% | 83.10% | 42.96% |
| FID (↓) | 16.70 | 19.34 | 16.86 | 19.27 | 23.01 | 27.40 | 20.63 | 24.47 |
| CLIP (↑) | 0.311 | 0.290 | 0.311 | 0.289 | 0.312 | 0.295 | 0.314 | 0.298 |

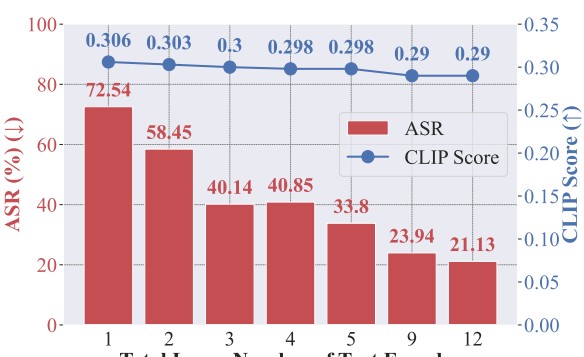

Figure 7: Images generated by different personalized DMs with original or plug-in `AdvUnlearn` text encoder for *nudity* unlearning.

preserved when plugging the text encoder obtained from `AdvUnlearn` in SD v1.4 into other DMs (see 'Transfer' performance vs. 'Original' performance). This is most significant when transferring to SD v1.5 due to its similarity with SD v1.4. For dissimilar DMs like DreamShaper [109] and Protogen [110], the `AdvUnlearn`-acquired text encoder in SD v1.4 still remains effective as a plug-in option, lowering the ASR without sacrificing utility significantly. **Fig. 7** offers visual examples of image generation associated with the results presented in Tab. 8.

**The effect of text encoder layers on DM unlearning.** In **Fig. 8**, we show the ASR (robustness metric) and the CLIP score (a utility metric) of post-nudity unlearning against various choices of text encoder layers for optimization in `AdvUnlearn`. Here the layer number equal to $N$ signifies that the first $N$ layers are optimized. We observe that the robustness gain escalates as more layers are optimized. In particular, optimizing only the initial layers failed to provide adequate robustness for DM unlearning against adversarial attacks, contrary to findings in [102], where shallower encoder layers suffice for guiding DMs in

Figure 8: Performance of `AdvUnlearn` vs. text encoder layers to optimize in nudity unlearning.

image editing albeit in a *non-adversarial* context. Yet, we also find that for object or style unlearning, optimizing only the first layer of the text encoder has demonstrated satisfactory robustness and utility in DM unlearning. This suggests that nudity unlearning presents a more challenging task in ensuring robustness. Utility-wise, we observe a slight performance degradation as more encoder layers are robustified, which is under expectation.

**Choice of adversarial attacks.** In **Tab. 9**, we show the attack success rate (ASR), the robustness metric of post-nudity unlearning against various choices of adversarial prompt attacks: ① CCE (circumventing concept erasure) [111] utilizes textual inversion to generate universal adversarial attacks in the embedding space. By inverting an erased concept into a 'new' token embedding, learned from multiple images featuring the target concept, this embedding is then inserted into the target text prompt. ② PEZ (hard prompts made easy) [111] is to generate an entire text prompt for the target image by optimizing through the cosine similarity. ③ PH2P (prompting hard or hardly prompting) [112] is similar to PEZ but with different optimization objective. ④ UnlearnDiffAtk [23] has been used as the default method for generating attacks in this work. When the attack is based on discrete prompts (such as UnlearnDiffAtk, PEZ, and PH2P), our proposed method AdvUnlearn consistently achieves remarkable erasure performance and robustness. Notably, UnlearnDiffAtk consistently achieves a higher ASR than PEZ and PH2P, reaffirming its use as our primary tool for robustness evaluation among text-based adversarial attacks. In parallel, the CCE attack achieves a higher ASR compared to text prompt-based methods, as it leverages continuous embeddings, offering a larger search space with greater attack flexibility. This is not surprising as the textual inversion is engineered to learn a 'new' continuous token embedding, enabling the representation of objects not encountered during training.

Table 9: Robustness evaluation of AdvUnlearn in terms of ASR (attack success rate) using various attack generation methods in different unlearning tasks (Nudity, Style, Object). A lower ASR indicates better robustness.

| Attack Method | Nudity | Van Gogh | Church | Parachute | Tench | Garbage Truck |
|---|---|---|---|---|---|---|
| UnlearnDiffAtk | 21.13% | 2% | 6% | 14% | 4% | 8% |
| CCE | 39.44% | 28% | 36% | 48% | 24% | 44% |
| PEZ | 3.52% | 0% | 2% | 0% | 0% | 4% |
| PH2P | 5.63% | 0% | 4% | 0% | 2% | 6% |

**Other ablation studies.** In **Appx. F**, we demonstrate more ablation studies. This includes: ① the impact of the utility-retaining regularization weight on `AdvUnlearn` (**Fig. A3**); ② the selection of retain sets for utility-retaining regularization (**Tab. A3**); ③ the impact of adversarial prompting strategy for `AdvUnlearn` (**Tab. A4**); ④ the robustness of SD v1.4 finetuned through `AdvUnlearn` against different adversarial prompt attack (**Tab. A5**).

# 7 Conclusion

Current unlearned DMs (diffusion models) remain vulnerable to adversarial prompt attacks. Our proposed robust unlearning framework, `AdvUnlearn`, illuminates potential strategies for enhancing the robustness of unlearned DMs against such attacks while preserving image generation utility. Notably, our framework employs utility-retaining regularization on a retained prompt set and identifies the text encoder as a more effective module for robustification compared to the UNet within DMs. Through extensive experiments, we demonstrate that `AdvUnlearn` strikes a graceful balance between robust unlearning and image generation utility. Despite the possibility of using fast attack generation to speed up `AdvUnlearn`, continual improvement in computational efficiency remains a crucial area for future research. Limitations and broader impacts are further discussed in **Appx. G** and **Appx. H**.

# Acknowledgement

Y. Zhang, J. Jia, Y. Zhang, C. Fan, J. Liu, and S. Liu were supported by the National Science Foundation (NSF) CISE Core Program Award IIS-2207052, the NSF Cyber-Physical Systems (CPS) Award CNS-2235231, the NSF CAREER Award IIS-2338068, the Cisco Research Award, and the Amazon Research Award for AI in Information Security.

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

# Appendix

## A  System Prompt for LLM as a Judger

We utilize a Large Language Model (LLM) like GPT-4 to act as a judger to filter out prompts that are relevant to the concept to be unlearned. Since nudity is a global concept [19], even if 'nudity' is not mentioned in the prompt, the generated image conditioned on the prompt might still involve the 'nudity' concept. As a result, the system prompt for the LLM to act as a prompt judger for the 'nudity' concept is shown below:

> *"Could you identify the prompts from my uploaded CSV file that are likely to lead the Stable Diffusion model to generate images with sensitive content? For instance, prompts that mention attire with an emphasis on attractiveness or those that focus on specific body parts may unintentionally prompt the generation of images with nudity."*

Unlike 'nudity', objects are local concepts, so we only need to filter out the name of the object that needs to be unlearned directly in the prompt. Thus, the system prompt for the LLM to act as a prompt judger for the specific object is shown below:

> *"Could you analyze the prompts from my uploaded CSV file to identify those that are likely to direct the Stable Diffusion model to generate images containing '{OBJECT}' or any of its synonyms?"*

## B  Algorithm Details: `AdvUnlearn`

**Algorithm 1** outlines the steps of `AdvUnlearn`. Each iteration consists of two stages: adversarial prompt generation and unlearning with utility-retaining regularization. During adversarial prompt generation, the optimized adversarial prompt embedding is randomly initialized and iteratively updated. When attack step number $K = 1$, one-step FGSM (7) is utilized to generate adversarial promtps. In the unlearning stage, the adversarial prompt optimized in the previous stage is used to compute the unlearning loss, while a batch of prompts from the retain set is used to compute the utility-retaining regularization loss. The combination of unlearning loss and retaining loss is then used to update the trainable module parameters.

---

**Algorithm 1** `AdvUnlearn`: Defensive Unlearning with Adversarial Training for DMs

---

1: Given Iteration Number $I$, batch size $b$ of retaining prompts $c_{\text{retain}}$, regularization weight $\gamma$, learning rate $\beta$, adversarial step size $\alpha$, attack step number $K$, unlearning concept $c$, the DM to be unlearned $\boldsymbol{\theta}$, and the frozen original DM $\boldsymbol{\theta}_{\text{o}}$:

2: **for** $i = 1, 2, \ldots, I$ **do**

3:     ♦ **Adversarial prompt generation**
4:     Randomly initialize adversarial soft prompt embedding $\delta_0$
5:     **if** K = 1 **then**
6:         $\delta = \delta_0 - \alpha \cdot \text{sign}\left(\nabla_{\boldsymbol{\delta}} \ell_{\text{atk}}(\boldsymbol{\theta}_i, c + \delta_0)\right)$        ▷ Fast AT variant: *One*-Step FGSM (7)
7:     **else**
8:         Run $K$-Step attacks using (4)
9:     **end if**

10:     ◇ **Defensive unlearning with utility-retaining regularization**
11:     $\ell_{\text{u}}(\boldsymbol{\theta}_i, c + \delta_K) = \ell_{\text{ESD}}(\boldsymbol{\theta}_i, c + \delta_K) + \gamma \mathbb{E}_{\tilde{c} \sim \mathcal{C}_{\text{retain}}} \left[\|\epsilon_{\boldsymbol{\theta}_i}(\mathbf{x}_t|\tilde{c}) - \epsilon_{\boldsymbol{\theta}_{\text{o}}}(\mathbf{x}_t|\tilde{c})\|_2^2\right]$
12:     $\boldsymbol{\theta}_{i+1} = \boldsymbol{\theta}_i - \beta \nabla_{\boldsymbol{\theta}} \ell_{\text{u}}(\boldsymbol{\theta}_i, c + \delta_K)$

13: **end for**

---

## C  Attack Setup

Following the methodology of [23], we introduce prepended prompt perturbations using $N$ tokens, where $N = 5$ is designated for *nudity unlearning* and $N = 3$ for both *style* and *object unlearning*. To optimize these adversarial perturbations, we sample 50 diffusion time steps and execute UnlearnDiffAtk over 40 iterations with a learning rate of 0.01, using the AdamW optimizer. For the classification of generated images, we use NudeNet [113] for nudity unlearning, a ViT-base model [23] fine-tuned on the WikiArt dataset for style unlearning, and an ImageNet-pretrained ResNet-50 for object unlearning tasks.

## D  Other Results and Visualizations for *Nudity* Unlearning

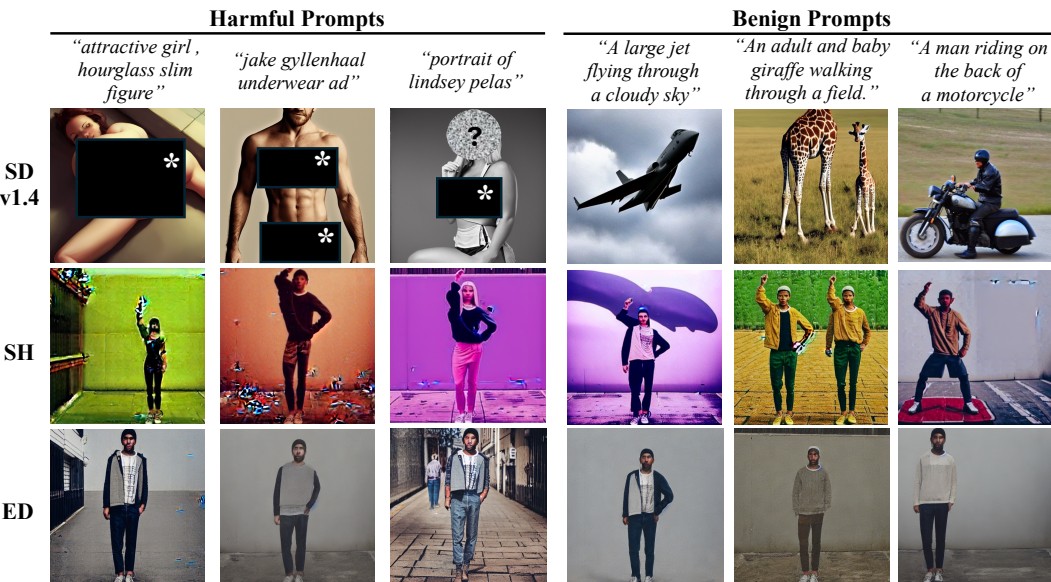

Figure A1: Visualization examples of images generated by SH and ED for *nudity* unlearning.

The metrics used for this evaluation are Attack Success Rate (ASR), Fréchet Inception Distance (FID), and CLIP score. As shown in **Tab. A1**, although ScissorHands (SH) [54] and EraseD­iff (ED) [53] achieve high unlearning robustness against adver­sarial prompt attacks, the trade-offs are significant. Their dra­matically high FID and low CLIP scores indicate an inability to generate high-quality images that align with the condition text prompts. This is further corroborated by their visualization ex­amples. In **Fig. A1**, we observe that, regardless of the condition prompts, the generated images are similar, demonstrating that

Table A1: Performance evaluation of additional unlearning methods, SH and ED, applied to the base SD v1.4 model for *nudity* unlearning.

| Metrics | SD v1.4 (Base) | SH | ED |
|---|---|---|---|
| ASR (↓) | 100% | 7.04% | 2.11% |
| FID (↓) | 16.70 | 128.53 | 233.31 |
| CLIP (↑) | 0.311 | 0.235 | 0.180 |

SH and ED fail to generate varied and contextually appropriate images. Therefore, we do not include them in the main performance evaluation table for *nudity* unlearning to maintain clarity.

# E Other Results and Visualizations for *Object* Unlearning

In **Tab. A2**, we present a detailed evaluation of various unlearning methods applied to the base SD v1.4 model for three different objects: *Garbage Truck*, *Parachute*, and *Tench*. The unlearning methods compared include FMN (Forget-Me-Not) [20], SPM (concept-

Table A2: Performance evaluation of unlearning methods applied to the base SD v1.4 model for *Garbage Truck*, *Parachute*, and *Tench* unlearning.

| Concept | Metrics | SD v1.4 (Base) | FMN | SPM | SalUn | ED | ESD | SH | AdvUnlearn (Ours) |
|---|---|---|---|---|---|---|---|---|---|
| **Garbage Truck** | ASR (↓) | 100% | 100% | 82% | 42% | 38% | 26% | 2% | 8% |
| | FID (↓) | 16.70 | 16.14 | 16.79 | 18.03 | 19.22 | 24.81 | 67.76 | 17.92 |
| | CLIP (↑) | 0.311 | 0.308 | 0.310 | 0.311 | 0.307 | 0.290 | 0.283 | 0.305 |
| **Parachute** | ASR (↓) | 100% | 100% | 96% | 74% | 82% | 58% | 24% | **14%** |
| | FID (↓) | 16.70 | 16.72 | 16.77 | 18.87 | 18.53 | 21.4 | 55.18 | 17.78 |
| | CLIP (↑) | 0.311 | 0.307 | 0.311 | 0.311 | 0.309 | 0.299 | 0.282 | 0.306 |
| **Tench** | ASR (↓) | 100% | 100% | 90% | 14% | 16% | 48% | 8% | **4%** |
| | FID (↓) | 16.70 | 16.45 | 16.75 | 17.97 | 17.13 | 18.12 | 57.66 | 17.26 |
| | CLIP (↑) | 0.311 | 0.308 | 0.311 | 0.313 | 0.310 | 0.301 | 0.280 | 0.307 |

SemiPermeable Membrane) [28], SalUn (saliency unlearning) [27], ED (EraseDiff) [53], ESD (erased stable diffusion) [19], SH (ScissorHands) [54], and our proposed DM unlearning scheme, referred to as AdvUnlearn. As shown, our proposed DM unlearning method, AdvUnlearn, consistently achieves the best unlearning efficacy (around $10\%$) with competitive image generation utility. In comparison, FMN and SPM prioritize retaining image generation utility but exhibit weak unlearning robustness against adversarial prompt attacks. Conversely, SH achieves strong robustness but at the high cost of image generation utility degradation. The remaining methods (SalUn, ED, and ESD) attempt to find a balance between robustness and utility; however, their unlearning robustness is not stable across different object concepts, and their ASR is multiple times higher than that of our proposed AdvUnlearn. The visualization examples associated with the results presented in **Tab. A2** can be found in **Fig. A2**. Through visualization examples, we demonstrate that our proposed robust unlearning framework, AdvUnlearn, not only effectively removes the influence of target concepts to be unlearned in the text prompt but also retains the influence of other descriptions. For instance, a garbage truck-unlearned DM equipped with the AdvUnlearn text encoder generates a photo of a parking lot from the text prompt 'garbage truck in a parking lot.' Similarly, other object-unlearned DMs with corresponding AdvUnlearn text encoders produce a photo of a desert landscape from the prompt 'parachute in a desert landscape' and a photo of a baby in a pond from the prompt 'baby tench in a pond.' Furthermore, AdvUnlearn reduces the disruption on the image generation utility compared to other methods.

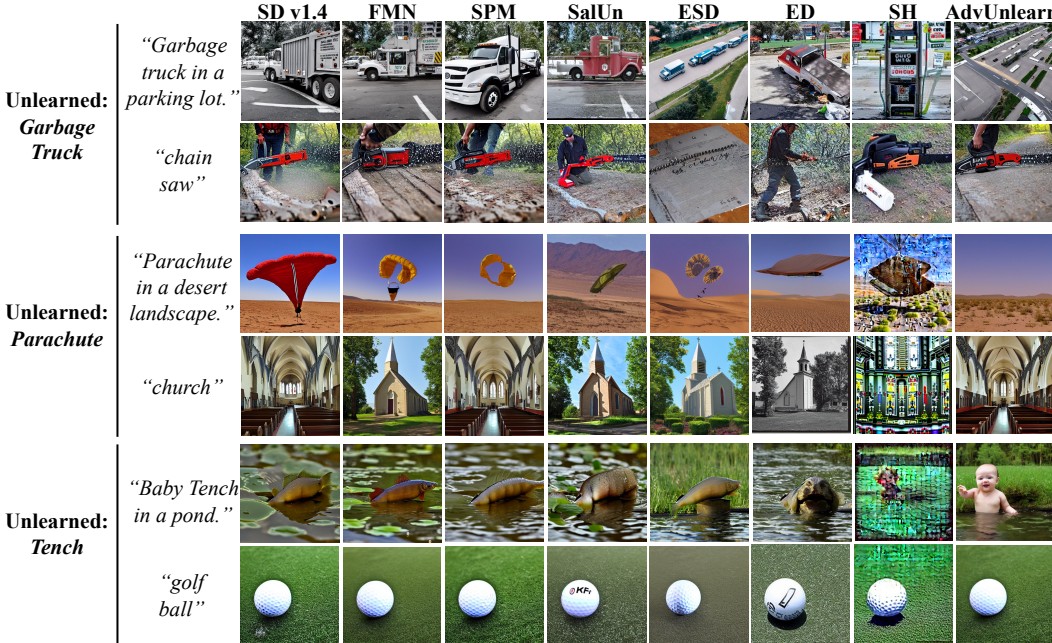

Figure A2: Visualization examples of images generated by different unlearning method for *Garbage Truck*, *Parachute*, and *Tench* unlearning.

# F    Other Ablation Studies

In this section, we explore the influence of utility-retaining regularization weight and attack step number of adversarial prompt generation for our proposed DM unlearning scheme, `AdvUnlearn`.

**Regularization weight.** As depicted in **Fig. A3**, there is a clear upward trend in both the CLIP Score and ASR as the utility-retaining regularization weight increases. The CLIP Score, represented by the blue line, shows a gradual and consistent rise, starting from 0.282 at a regularization weight of 0.1 and reaching 0.3 at a weight of 0.7, indicating improved image generation utility. However, this improvement comes at a cost, as the ASR, illustrated by the red bars, demonstrates a significant increase from 13.18% at a weight of 0.1 to 36.63% at a weight of 0.7, suggest-

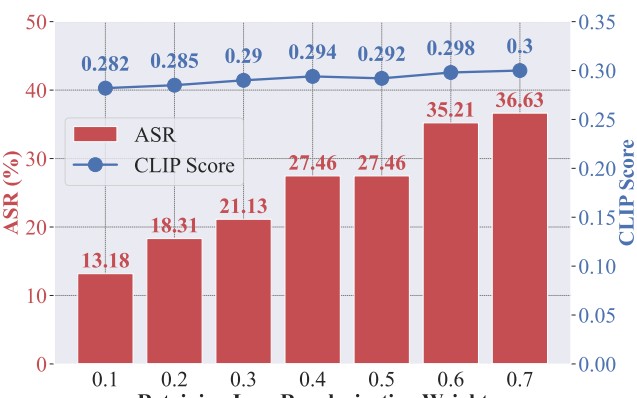

Figure A3: Performance comparison of different utility-retaining regularization weight for `AdvUnlearn`.

ing a degradation in unlearning efficacy. Consequently, we have selected a regularization weight of 0.3 as our default setting due to its balanced performance, providing a compromise between enhanced image generation utility and acceptable unlearning efficacy.

**Retain set selection.** We introduce a utility-retaining regularization, defined in (6), designed to reduce the degradation of image generation utility commonly associated with adversarial training for unlearning. In **Tab. A3**, we examine the influence of object class sources on the retain set, using the template *'a photo of [OBJECT]'*, and evaluate the effectiveness of employing a Large Language Model (LLM) as a

Table A3: Performance evaluations of `AdvUnlearn` using different retaining prompt dataset for *nudity* unlearning.

| Source: | | ImageNet | | COCO Object | |
|---|---|---|---|---|---|
| **LLM Prompt Filter:** | | ✘ | ✔ | ✘ | ✔ |
| ASR (↓) | | 51.41% | 45.07% | 38.03% | **21.13**% |
| FID (↓) | | 19.09 | 19.10 | 18.85 | 19.34 |
| CLIP (↑) | | 0.301 | 0.299 | 0.293 | 0.290 |

prompt filter, which helps exclude prompts potentially related to the concept being erased. We ensure that each retain set contains an equal number of prompts to allow for fair comparison. Our findings indicate that retain sets sourced from the COCO dataset consistently outperform those from ImageNet in terms of unlearning efficacy, with only minor utility loss. Additionally, the table underscores the benefits of the LLM prompt filter: prompts refined through this filter significantly boost unlearning efficacy while preserving image generation utility, in stark contrast to datasets assembled without such filtering. Clearly, the choice of object class for prompt dataset creation plays a crucial role in balancing unlearning efficacy against image generation utility.

**Adversarial prompting strategy.** We evaluate three distinct adversarial prompting strategies for adversarial prompt generation in `AdvUnlearn`: ① *Replace*: This strategy involves directly replacing the original concept prompt with an optimized adversarial soft prompt. ② *Add*: This method adds the optimized adversarial soft prompt to the original concept prompt within the token embedding space. ③ *Prefix*: This approach prepends

Table A4: Performance evaluations of `AdvUnlearn` using various adversarial prompting for *nudity* unlearning.

| Adversarial Prompting: | | Replace | Add | Prefix |
|---|---|---|---|---|
| | ASR (↓) | 36.63% | 45.07% | **21.13**% |
| **Metrics:** | FID (↓) | 19.39 | 19.60 | 19.34 |
| | CLIP (↑) | 0.298 | 0.299 | 0.290 |

the optimized adversarial soft prompt before the original concept prompt, and is the default setting for our study. As demonstrated in **Tab. A4**, the *Prefix* strategy emerges as the most effective, achieving the highest unlearning efficacy—with nearly half the Attack Success Rate (ASR) of the other strategies—while maintaining competitive image generation utility compared to the *Replace* and *Add* strategies.

**Robustness gain from `AdvUnlearn` at different test-time attacks.** In Tab. A5, we present the robustness of SD v1.4 post-nudity unlearning when facing different test-time adversarial prompt attacks, UnlearnDiffAtk [23] (default choice) and P4D [24]. As we

Table A5: Robustness of SD v1.4 post-nudity unlearning against different test-time adversarial attacks.

| Metrics | FMN | SPM | UCE | ESD | SalUn | `AdvUnlearn` (Ours) |
|---|---|---|---|---|---|---|
| P4D - ASR ($\downarrow$) | 97.89% | 91.55% | 75.35% | 71.27% | 12.68% | 19.72% |
| UnlearnDiffAtk - ASR ($\downarrow$) | 97.89% | 91.55% | 79.58% | 73.24% | 11.27% | 21.13% |

can see, `AdvUnlearn` maintains its effectiveness in improving robustness under both attack methods at testing time. Notably, the ASR against P4D is even lower than that against UnlearnDiffAtk. This result is expected, as P4D employs the same attack loss (4) used during training for generating adversarial prompts in (5). Therefore, we default to UnlearnDiffAtk for test-time attacks in robustness evaluation due to its unseen nature during training.

## G    Limitations

This work seeks to improve the robustness of concept erasing by incorporating the principles of adversarial training (AT) into the process of machine unlearning, resulting in a robust unlearning framework named AdvUnlearn. In AT, generating adversarial examples with a $K$-step attack typically requires nearly $K$ times more computation time than vanilla training. Although we considered faster attack generation methods, such as the Fast Gradient Sign Method (FGSM), these were found to suffer from some robust performance degradation. Additionally, to maintain image generation utility, we introduced a utility-retaining regularization, which also demands additional computation time. Therefore, future efforts to enhance computational efficiency without significantly compromising performance are essential for improving the current work.

## H    Broader Impacts

The broader impacts of this study include social and ethical implications, where improved reliability of concept erasing aligns AI technologies with societal norms and ethical standards, potentially reducing the spread of harmful digital content. Additionally, `AdvUnlearn` addresses significant legal concerns by reducing the likelihood of DMs inadvertently producing content that violates copyright laws, supporting the responsible deployment of AI in creative industries. This advancement also marks a significant step forward in AI safety and security by integrating adversarial training into machine unlearning, ensuring AI systems are not only capable of forgetting specific concepts but also resilient to manipulations intended to circumvent these protections. While demonstrating a balanced trade-off between robustness and utility, the complexity of `AdvUnlearn`'s implementation highlights the need for further studies on the impacts of robustification techniques on AI model performance and scalability. Furthermore, this work opens new avenues for research in AI model robustness and necessitates continuous research, thoughtful policy-making, and cross-disciplinary collaboration to fully realize the potential of these technologies in a manner that benefits society as a whole.

