# OpenReview forum: "Defensive Unlearning with Adversarial Training for Robust Concept Erasure in Diffusion Models"
_NeurIPS.cc/2024/Conference — NeurIPS 2024 poster_

### Official Review · Reviewer_hd8m · 2024-07-12

**Soundness:** 3
**Presentation:** 3
**Contribution:** 3
**Rating:** 5
**Confidence:** 4

**Summary:**

This paper introduces AdvUnlearn, a robust unlearning framework that integrates adversarial training into diffusion models to enhance concept erasure. This method aims to prevent DMs from generating harmful or inappropriate content, such as nudity, even under adversarial prompt attacks. AdvUnlearn focuses on optimizing the text encoder rather than the UNet, achieving a balance between effective concept erasure and high image generation quality. Extensive experiments demonstrate AdvUnlearn's robustness and utility across various unlearning scenarios.

**Strengths:**

1. Enhanced Robustness:
AdvUnlearn improves the robustness of diffusion models against adversarial prompt attacks, effectively preventing the generation of undesired content.

2. Maintains Image Quality:
By focusing on the text encoder and utilizing utility-retaining regularization, AdvUnlearn preserves high image generation quality post-unlearning.

3. Plug-and-Play Usability:
The method allows the robust text encoder to be shared across different DMs, making it easy to implement and enhancing its practical usability.

**Weaknesses:**

1. ASR needs to be measured based on more baseline attacks. Since the proposed method is based on the CLIP text encoder, incorporating CLIP-based adversarial attack methods [1] and [2] would provide greater understanding.


[1] Wen et al., Hard Prompts Made Easy: Gradient-Based Discrete Optimization for Prompt Tuning and Discovery\
[2] Mahajan et al., Prompting Hard or Hardly Prompting: Prompt Inversion for Text-to-Image Diffusion Models

**Questions:**

1. I thoroughly reviewed the algorithm in the Appendix. However, I need more details on how the text encoder is optimized using Eq.(5). The current form of Eq.(5) does not seem to include the text encoder. Please provide more details on this process.

**Limitations:**

1. More discussion comparing the proposed method with CLIP-based attacks [1] and [2] would be welcomed.

[1] Wen et al., "Hard Prompts Made Easy: Gradient-Based Discrete Optimization for Prompt Tuning and Discovery"\
[2] Mahajan et al., "Prompting Hard or Hardly Prompting: Prompt Inversion for Text-to-Image Diffusion Models"

---

> ### Author Rebuttal · Authors · 2024-08-07
>
> **Tables  (referred to as Table Rx)  can be found in the [attached PDF file](https://openreview.net/attachment?id=dQxPvBUICW&name=pdf).**
>
> **W1 & Q2 & Limitations: More attacks for ASR measure.**
>
> **A**: Thank you for the suggestions regarding two additional related works. We will cite them and expand our discussions in the related work section accordingly. Furthermore, following your recommendation, we have employed both the PEZ [1] and PHP2 [2] methods to evaluate our proposed method, AdvUnlearn, as detailed in **Table R1**. The results demonstrate that UnlearnDiffAtk consistently achieves a higher Attack Success Rate (ASR) compared to PEZ and PHP2, reaffirming its effectiveness as our primary tool for robustness evaluation among text-based adversarial attacks. **More details and analysis can be found in [General Response 1 (GR1)](https://openreview.net/forum?id=dkpmfIydrF&noteId=dQxPvBUICW).**
>
> In the revision, we will add a detailed comparison with the PEZ [1] and PHP2 [2] attacks and their discussion.
>
> **Q1: How is text encoder to be involved in optimization?**
>
> **A**: Sorry for the confusion. Eq. (5) outlines a generic bi-level optimization problem formulation used in AdvUnlearn, where $\boldsymbol \theta$  represents the upper-level optimization variables in a general sense. In Algorithm 1, $\boldsymbol \theta$  specifically refers to the parameters of the text encoder. Alternatively, $\boldsymbol \theta$  can be considered as a vector that includes the text encoder parameters to be optimized and the UNet parameters that remain frozen during the optimization process. Steps 10-12 in the algorithm are applied exclusively to the active text encoder parameters, ensuring that adjustments are made only where necessary for effective unlearning.
>
>
>
> > [1] Hard Prompts Made Easy: Gradient-Based Discrete Optimization for Prompt Tuning and Discovery, NeurIPS 2023.
>
> > [2] Prompting Hard or Hardly Prompting: Prompt Inversion for Text-to-Image Diffusion Model, CVPR 2024.

---

> > ### Comment · Reviewer_hd8m · 2024-08-12
> >
> > Thank you for your clarification and for providing the additional experimental results. However, I believe further clarification is still needed. Specifically, I noticed that Table 1 of the additional PDF does not include results from the naive model for comparison. To make a proper comparison, it is essential to show the ASR of PEZ and PH2P first. After that, reporting the robustness would provide a clearer understanding. Additionally, it seems evident that UnlearnDiff has a higher success rate than PEZ, as PEZ is more akin to a black-box attack in terms of access to the U-Net.
> >
> > Given these points, I will maintain my score. While the paper presents good motivation, it requires revisions to address the questions raised by other reviewers and myself.

---

> ### Author Response · Authors · 2024-08-13
> **Gratitude for Your Positive Feedback and Continued Discussion**
>
> Dear Reviewer hd8m,
>
> Thank you for recognizing our rebuttal efforts and for your continued engagement with our submission, particularly your request for further clarification. We provide our further response below.
>
> We have now included the ASR for the original base model (SD v1.4) across different attack methods in **Table R8** of our response. As indicated, UnlearnDiffAtk demonstrates higher ASRs compared to PEZ and PH2P, as well as CCE (see GR1). We acknowledge your correct assessment that PEZ, operating more like a black-box attack, leverages only the text encoder within the DM, thus exhibiting a weaker attack profile. In contrast, UnlearnDiffAtk, designed specifically to test the robustness of concept erasing/unlearning in DMs, shows a significantly stronger attack performance than other attack methods. This is also noted in our paper (Tables 5-7), where we showed a 100% ASR for UnlearnDiffAtk against the original DM to underscore its effectiveness as an effective attack method in evaluation.  This high ASR reflects UnlearnDiffAtk's tailored approach as a white-box, text-based adversarial prompt method, which is valid for a rigorous assessment of unlearned models.
>
> In light of your feedback, we will include these experimental justifications in the revised manuscript to address concerns regarding the selection of UnlearnDiffAtk and its comparative analysis with other prompt-level attack methods. Although these revisions strengthen our paper, they do not deviate from the original contributions, e.g., the validity of using UnlearnDiffAtk in evaluation.
>
> **Table R8: Attack evaluation of **base model** (w/o applying AdvUnlearn) in ASR across various generation concepts of our interest (Nudity, Style, Object)  under distinct attacks. A higher ASR indicates stronger  attack performance.**
> |                           | Nudity | Van Gogh | Church | Parachute | Tench | Garbage Truck |
> |-------------------------------|--------|----------|--------|-----------|-------|---------------|
> | ASR (Under UnlearnDiffAtk)    | 100%   | 100%     | 100%   | 100%      | 100%  | 100%          |
> | ASR (Under CCE Attack)        | 82.39% | 90%      | 88%    | 96%       | 80%   | 88%           |
> | ASR (Under PEZ Attack)        | 45.07% | 34%      | 48%    | 52%       | 42%   | 46%           |
> | ASR (Under PH2P Attack)       | 54.93% | 42%      | 62%    | 58%       | 48%   | 56%           |
>
> We hope the above response addresses your remaining concerns adequately. If you have any more questions or need further discussion, please feel free to reach out. We will try our best to ensure that all your concerns can be thoroughly addressed before the rebuttal period concludes.
>
> Thank you very much,
>
> Authors

---

### Official Review · Reviewer_4rR4 · 2024-07-12

**Soundness:** 2
**Presentation:** 3
**Contribution:** 3
**Rating:** 5
**Confidence:** 5

**Summary:**

The authors propose a method for robust (against adversarial attacks) concept-erasing for latent diffusion models. Specifically text-to-image diffusion models.

The main contributions are:
1. Integrating adversarial training into machine unlearning by modifying it as a bi-level optimization problem.
2. Contrary to the conventional techniques where only the parameters of the UNeT is updated, the authors show that updating the text encoder can effectively maintains a robustness-utility trade-off.

**Strengths:**

1. The problem of robust-concept erasing is relatively new and important to handle.
2. The authors have formulated the task clearly and proposed a solution to address this important problem.
3. The initial results on text-based attacks (UnlearnDiffAttack) demonstrate an interesting direction.

**Weaknesses:**

Lack of Sufficient Evaluation:
1. The authors majorly evaluate the robustness of the method against only the UnlearnDiff attack, which is a text-based attack.
2. They have not extended their evaluations to other recent peer-reviewed SOTA attacks such as the CCE (ICLR-24) [1] and RAB (ICLR-24) [2] attacks.
3. Specifically, CCE is a strong inversion-based attack that introduces a new token into the vocabulary. CCE relatively takes much less time than the UnlearnDiffAttack.
4. GIVEN CCE ATTACK INVERTS THE ERASED CONCEPT INTO A NEW TOKEN, THE AUTHOR'S CLAIMS OF UPDATING ONLY THE TEXT-ENCODER FOR A BETTER ROBUSTNESS-UTILITY TRADEOFF CAN BE CALLED INTO QUESTION.
5. In addition to the UnlearnDiffAttack, the performance of AdvUnlearn on CCE would be the right benchmark to assess the model's robustness.

Lack of Completeness:
1. The authors have not presented the quantitative erasing results of AdvUnlearn for any of the unlearning scenarios. In addition to the robustness and utility performance, studying the erasing performance of the method is important to understand whether optimizing for the former two properties has any compromise on the latter's performance.

References:
[1] https://arxiv.org/pdf/2308.01508 (Circumventing Concept Erasure Methods For
Text-to-Image Generative Models)
[2] https://arxiv.org/pdf/2310.10012 (RING-A-BELL! HOW RELIABLE ARE CONCEPT REMOVAL METHODS FOR DIFFUSION MODELS?)

**Questions:**

Nudity Unlearning:
1. Why have the authors chosen only a subset of the 4703 I2P prompts?
2. How many prompts are there in the subset?
2. On what filters was the subset chosen?
3. Please elaborate on why the erasing results on the original I2P benchmark and the nudity count are missing.

Style Unlearning:
1. Please elaborate on why the erasing results for the art unlearning scenario is not discussed.
2. The authors have not discussed how erasing the style of one artist affects the other artistic styles.

Objects Unlearning:
1. Please elaborate on why the erasing results of each of the object scenarios are not discussed.
2. For the object unlearning scenarios, the experimental setup of evaluating against only 50 prompts seems limited. Baseline method such as ESD evaluate on a large set of 500 samples and also present the results on the remaining classes to understand the loss of utility. This analysis is missing here.

Utility:
1. Why have the authors chosen to compute the FID and CLIP scores on the subset of 10K prompts and not the standard 30K prompts like previous methods in the benchmark?
2. Why do the FID and CLIP scores of AdvUnlearn vary significantly for different concept/unlearning scenarios? eg: FID/CLIP score is 19.34/0.29 for nudity-unlearning and 16.96/0.30 for style-unlearning? On the other hand, the performance of baseline ESD does not vary as significantly: 18.18/0.30 (nudity) vs 18.71/0.30 (style).

**Limitations:**

1. The authors have discussed the limitations of the proposed method.
2. However, as highlighted in the "weakness" section, the author's claim of robustification through updating only the text-encoder cannot be verified unless evaluated against other adversarial attacks from the literature such as CCE and RAB.

---

> ### Author Rebuttal · Authors · 2024-08-07
>
> **Tables can be found in the attached PDF file.**
>
> **W1: Lack of Sufficient Evaluation across various attacks**
>
> **A**: Thank you for your valuable feedback concerning the scope of our evaluation.  We have included more attacks (including CCE and RAB) for evaluation. **Details can be found in General Response 1 (GR1).**
>
>
> **W2: Lack of Completeness: Missing quantitative erasure results of AdvUnlearn.**
>
> **A**: Following the reviewer’s suggestion, we have included additional experiments in Table R3 in the attached PDF file. **More detailed analysis can be found in General Response 2 (GR2).**
>
>
> **Q1: More explanation about prompt subset creation for Nudity unlearning.**
>
> **A**: In alignment with the established UnlearnDiffAtk protocol, we used the subset of 142 prompts selected from the I2P dataset. These prompts were specifically chosen based on the NudeNet score of the images they generated, where only those prompts generating images with a score higher than 0.75 were included. This threshold was selected to ensure that the prompts represent the most explicit and potentially harmful cases for nudity generation. We continue using this subset because it facilitates us to evaluate the robustness of our method against the worst-case prompt set (characterized by high nudity levels). Additionally, given the substantial computational costs associated with UnlearnDiff evaluations, focusing on this subset characterized by high nudity levels enables a more efficient use of resources while still rigorously assessing the performance of multiple models under challenging conditions.
>
> **Q2: Impact of style unlearning on other art styles.**
>
> **A**:  Based on your suggestion, we have conducted further experiments to specifically assess the impact of erasing Van Gogh's style on the generation of images styled after Monet, Paul Cezanne, and Andy Warhol. These experiments, detailed in **Table R4**, use a style classifier similar to that employed in our attack evaluations. Higher accuracy in this context suggests that the unlearning process has less impact on other artistic styles. Our results indicate that the style of Monet, who shared the Impressionist movement with Van Gogh, exhibited the most significant influence. In contrast, the styles of artists that more loosely relate to Van Gogh, such as Cezanne and Warhol, showed minimal impact. This suggests that unlearning tends to have side effects primarily on concepts closely related to the target of the unlearning process. These observations align with recent findings on the retainability of concept erasing in the recent study UnlearnCanvas [1]. We will include these additional experimental results and discussions in our revised manuscript. Thank you for the insightful comment.
>
> **Q3: Missing erasure performance and its effects on object generation utility in object unlearning, with considerations on evaluation prompt set size.**
>
> **A**: The omission of detailed erasing results for unperturbed forgetting prompts follows the rationale previously outlined in our response to W2. For further detail, please refer to the additional erasing results in Table R3, which include outcomes for objects such as Church, Garbage Truck, Parachute, and Tench (used in Figure 6 and Appendix E).
>
> Responding to another of your suggestions, we have expanded our robustness evaluation to include a larger set of 150 prompts, as presented in **Table R5**. The results from this expanded set align with our initial findings using 50 prompts, confirming the consistency of our method's performance. We are open to further expanding our prompt set to 500, akin to the baseline method ESD, when we can get more computing resources in the revision.
>
> Regarding the assessment of utility loss in our study, we utilized both FID and CLIP scores (Lines 345-346) to evaluate utility across various unlearning tasks. As shown in Table 7, there is a noticeable reduction in utility across different unlearning methods for object unlearning scenarios. However, AdvUnlearn demonstrates the favorable balance between maintaining utility and achieving robustness. Additionally, we conducted a comparative analysis of AdvUnlearn and ESD focusing on their image generation accuracy for both the targeted forgetting class and the remaining classes (500 images per class). The results, presented in **Table R6**, indicate that AdvUnlearn outperforms ESD in preserving utility for the remaining classes. This is attributed to the inclusion of utility regularization in our method, which significantly enhances the utility-unlearning tradeoff, as shown in Table 7.
>
> **Q4: Considerations on evaluation prompt set size for utility performance evaluation and higher utility performance variances for AdvUnlearn.**
>
> **A**: We evaluated models across different unlearning tasks using datasets of 10k and 30k prompts and found that they achieved similar results. As indicated in **Table R7**, the variance in FID and CLIP scores for the AdvUnlearn models is small. Consequently, we decided to utilize 10k prompts for evaluations to moderate computational requirements, ensuring efficiency without compromising the integrity of our results.
>
> We acknowledge that as the unlearning task varies, there is a higher variance in FID and CLIP scores for AdvUnlearn compared to ESD, which can be attributed to differences in the "optimization variables" (i.e., the specific modules optimized during unlearning) between these two methods. As illustrated in Figure 8, unlearning global knowledge concepts such as nudity requires tuning the entire text encoder to achieve the best unlearning performance. This involvement of more layers tends to result in more utility degradation. Conversely, as noted in Lines 452-454, specific styles and objects represent more localized concepts. This allows for the tuning of only the first layer of the text encoder, which is sufficient to unlearn while preserving better overall utility.
>
> > [1] Unlearncanvas…., Arxiv.

---

> > ### Comment · Reviewer_4rR4 · 2024-08-12
> >
> > I appreciate the author's detailed rebuttal.
> > The authors have clarified all my queries and I am willing to increase my rating.

---

> > > ### Author Response · Authors · 2024-08-13
> > > **Gratitude for Your Positive Feedback and Willingness to Increase Rating**
> > >
> > > Dear Reviewer 4rR4,
> > >
> > > We are delighted to learn that our rebuttal has **successfully addressed all your questions and concerns.** We are committed to further enhancing our paper based on your insightful feedback.
> > >
> > > Additionally, we are very grateful for **your willingness to consider increasing the original rating (a score of 4) to a higher rating.** It would be highly appreciated if you could make this adjustment during the author-reviewer discussion phase. Your support and acknowledgment of the efforts we have put into our rebuttal are incredibly encouraging.
> > >
> > > Thank you!
> > >
> > > Authors

---

### Official Review · Reviewer_gHPo · 2024-07-12

**Soundness:** 3
**Presentation:** 3
**Contribution:** 3
**Rating:** 5
**Confidence:** 2

**Summary:**

This paper proposes AdvUnlearn, a method aimed at enhancing the robustness of concept erasing. The approach integrates the principles of adversarial training into the unlearning process. Specifically, the text encoder is identified as the most suitable module for robustification. Experiments are conducted on eight DM unlearning methods, including nudity unlearning, style unlearning, and object unlearning.

**Strengths:**

1, The problem addressed in this paper is significant. Enhancing the robustness of concept erasing could substantially reduce safety risks and copyright violations.

2, The idea is intuitive and reasonable. Adversarial training can be used to enhance robustness, and the paper identifies a straightforward method to preserve image quality while focusing on the text encoder as the most suitable module for robustification.

3, The experiments are relatively thorough, including eight unlearning baselines and various concept-erasing tasks.

**Weaknesses:**

The experimental section lacks a sufficient number of attack methods.

The introduction to C_{retrain} is inadequate.

**Questions:**

The key questions preventing me from increasing my rating are:

I don't understand why only UnlearnDiffAtk is used for robustness evaluation. Since the robustification targets the text encoder, it seems that the textual inversion-based attack [65] should also be used for robustness evaluation. Additionally, I suggest that more reasonable attacks be considered for robustness evaluation.

I also question the approach of mainly robustifying the text encoder. Even after adversarial training, I believe there will still be an embedding that can generate the target concept. This defense could be easily bypassed by some embedding-based attack.

I don't understand how C_{retrain} is selected to retain model utility. Are the concepts not covered in C_{retrain} still suffering from degradation?

**Limitations:**

The limitations have been discussed in the appendix.

---

> ### Author Rebuttal · Authors · 2024-08-07
>
> **Tables  (referred to as Table Rx)  can be found in the [attached PDF file](https://openreview.net/attachment?id=dQxPvBUICW&name=pdf).**
>
> **W1 & Q1: Lacks a sufficient number of attack methods.**
>
> **A**: Thank you for your insightful suggestion regarding the range of attack methods in our study. In response, we have incorporated three additional attack evaluation methods, as detailed in the General Response. Among all the text prompt-based attacks we assessed, UnlearnDiffAtk continues to stand out as the most effective, reaffirming its selection for our initial analysis. **For a more detailed evaluation and comparison of these methods, please refer to the [General Response 1 (GR1)](https://openreview.net/forum?id=dkpmfIydrF&noteId=dQxPvBUICW).**
>
>
> **W2 & Q3: Introduction to $C_{retain}$ is inadequate.**
>
> **A**: Thank you for pointing out the need for clearer information regarding the construction of  $C_{retain}$. First, we described the composition of the retaining dataset in Lines 246-249 and Appendix A. Specifically, the prompts in $C_{retain}$ are formatted as "a photo of [OBJECT CLASS]," where "OBJECT CLASS" is sampled from well-known object datasets such as ImageNet or COCO. These prompts undergo a filtering process using a large language model (LLM) to ensure they exclude any concepts targeted for erasure, maintaining their suitability as retain prompts (Appendix A). The finalized $C_{retain}$ consists of  243 distinct prompts. During training, 5  prompts from $C_{retain}$​ are randomly selected for each batch to support utility-retaining regularization. It's important to clarify that the role of $C_{retain}$​ is not to train the model on producing specific objects or concepts; instead, it aims to guide the model in generating general, non-forgetting content effectively. This approach helps counteract the potential performance drops often seen with adversarial training [1] (Lines 236-238). As evidenced in **Fig. 4-6** of the submission, incorporating $C_{retain}$​ enhances the general utility of the unlearned DM during the testing phase. In these figures, test-time prompts include varied objects like "toilet," "Picasso," and "cassette player" which are not part of $C_{retain}$, demonstrating the unlearned model's generalization capabilities.
>
>
>
> **Q2: Why not consider embedding-based attacks?**
>
> **A**: Thank you for raising this question about the choice of attack method. We opted for UnlearnDiffAtack as our primary attack evaluation method for the following reasons:
>
> - Our proposed defense, AdvUnlearn, is designed to fortify unlearned Diffusion Models (DMs) against text-based attacks post-unlearning in Eq. (5). Similar to how PGD attacks serve as a benchmark for evaluating adversarial training, we selected a prompt-based attack to assess the robustness of our unlearning approach.
> - To the best of our knowledge, UnlearnDiffAtack is a state-of-the-art, prompt-based text adversarial attack specifically designed to challenge unlearned DMs. It has demonstrated superior Attack Success Rate (ASR) compared to other text-based prompt inversion methods such as PEZ [2] and PH2P [3], as validated in our additional experiments **detailed in the General Response 1 (GR1) and Table R1**.
>
> However, we recognize the value of exploring diverse types of attacks, including embedding-based attacks such as CCE [4]. The CCE attack, which utilizes textual inversion to embed adversarial content, indeed shows higher ASR than discrete text-based attacks, as shown in Table R1. This is expected as our defense, AdvUnlearn, primarily addresses discrete prompt-based attacks. The continuous search space offered by CCE provides greater optimization flexibility, potentially allowing it to bypass certain defensive mechanisms.
>
> Acknowledging that the arms race between attack and defense is ongoing, similar to adversarial scenarios in discriminative models, we are committed to including an evaluation against the CCE attack in our revised paper and clarify the above points regarding our choice of UnlearnDiffAtack.
>
> > [1] Unlabeled data improves adversarial robustness, NeurIPS 2019.
>
> > [2] Hard Prompts Made Easy: Gradient-Based Discrete Optimization for Prompt Tuning and Discovery, NeurIPS 2023.
>
> > [3] Prompting Hard or Hardly Prompting: Prompt Inversion for Text-to-Image Diffusion Model, CVPR 2024.
>
> > [4] Circumventing Concept Erasure Methods For Text-to-Image Generative Model, ICLR 2023.

---

> > ### Comment · Reviewer_gHPo · 2024-08-11
> >
> > I appreciate the authors' detailed rebuttal and additional insights. Most of my questions have been addressed, and I will be maintaining my original rating.

---

> > > ### Author Response · Authors · 2024-08-13
> > > **Gratitude for Your Positive Feedback and Continued Discussion**
> > >
> > > Dear Reviewer gHPo,
> > >
> > > Thank you for your recognition of our detailed rebuttal and for maintaining your positive assessment with a score of 5. We are glad to hear that our responses have successfully addressed **most of your questions**.
> > >
> > > Based on your insightful comments regarding "The key questions preventing me from increasing my rating..", we sincerely hope that our detailed responses have adequately addressed these questions and helped to reinforce your confidence in our work. If you believe our rebuttal efforts in addressing these questions are meritorious, we would greatly appreciate it if you could consider increasing your rating.
> > >
> > > Should there be any more questions or need for further discussion, please feel free to reach out. We are fully prepared to continue our dialogue to ensure that all aspects of your concerns are thoroughly addressed before the rebuttal period concludes.
> > >
> > > Thank you once again for your thoughtful feedback and consideration.
> > >
> > >
> > > Warm regards,
> > >
> > > Authors

---

### Author Rebuttal · Authors · 2024-08-07

Thank you all for the thoughtful reviews and suggestions provided. In response, we will meticulously address each raised question and concern sequentially. **We choose to add the additional experiments via tables (referred to as Table Rx) in the [attached PDF file](https://openreview.net/attachment?id=dQxPvBUICW&name=pdf).**

We begin by general responses (GRs) in the following:

**GR1: Various attacks for robustness evaluation. (@hd8m, 4rR4, gHPo)**

**A**: Following this guidance, we have included three more  attack evaluation methods below:
- **CCE** (Circumventing Concept Erasure) [1]: The method employs textual inversion to generate universal adversarial attacks in the continuous embedding space. By inverting an erased concept into ***a 'new' token embedding***, learned from multiple images featuring the target concept, this embedding is then inserted into the target text prompt. This guides the diffusion model to generate images that contain the target concepts.

- **PEZ** (Hard Prompts made EaZ) [2]: The method generates ***an entire text prompt*** for the target image by optimizing through the cosine similarity between the outputs of the text encoder and the image encoder in the discrete space.

- **PH2P** (Prompting Hard or Hardly Prompting) [3]: Similar to PEZ, this method generates ***an entire text prompt*** for each target image. However, it employs a different optimization objective: minimizing the denoising error of the Latent Diffusion Model (LDM).

Given that the interface for text-to-image generation typically operates on text inputs, current attacks focus predominantly on hard prompts in discrete spaces. **Table R1** illustrates that when the attack is based on discrete prompts—a common scenario—our proposed method consistently achieves remarkable erasure performance and robustness. Notably, UnlearnDiffAtk consistently achieves a higher Attack Success Rate (ASR) than PEZ and PH2P, reaffirming its use as our primary tool for robustness evaluation among text-based adversarial attacks. In parallel, the CCE attack also records higher ASR than text-prompt based methods in Table R1, as it exploits continuous embeddings—providing a search space with greater freedom and easier optimization capabilities for attack generation. This is expected, given that our defense primarily targets discrete prompt-based attacks. **In the revision we will include the above additional evaluation methods and the discussion to provide a more nuanced analysis of our defense's effectiveness against different attack strategies.**

**(Additional RAB attack evaluation @Reviewer 4rR4)** We also responded to the suggestion for a broader evaluation by testing the RAB attack [4], which focuses on NSFW unlearning, with varying token lengths, presented in **Table R2**. Our method exhibited consistent robustness against RAB across different token lengths, further validating the effectiveness of AdvUnlearn. Additionally, the token lengths of 38 and 77 are not necessary for the nudity unlearning scenario, which aligns with the insights presented in the RAB paper.

Last but not the least, we would like to further clarify our reasons for choosing UnlearnDiffAtack as the attack evaluation method in our submission.
- **Our proposed defense, AdvUnlearn, is designed to fortify unlearned DMs against text-based attacks post-unlearning in Eq. (5).** Similar to how PGD attacks serve as a benchmark for evaluating adversarial training, we selected a prompt-based attack to assess the robustness of our unlearning approach.
- To the best of our knowledge, UnlearnDiffAtack is the state-of-the-art, prompt-based text adversarial attack specifically designed to challenge unlearned DMs. It has demonstrated better attack performance than PEZ [2] and PH2P [3], as validated above.

**GR2: Erasure performances. (@4rR4)**

**A**: Following the reviewer’s suggestion, we have included additional experiment results in **Table R3**, which specifically measures the effectiveness of AdvUnlearn in erasing unwanted content from outputs generated in response to non-attacked inappropriate prompts related to the targeted forgetting concept. In these tests, AdvUnlearn demonstrates robust performance in preventing the generation of content from inappropriate prompts, even when not under adversarial attack conditions. This is compared to other baseline methods, such as ESD, which shows slightly worse performance in similar tests. The effectiveness of AdvUnlearn in handling unperturbed inappropriate prompts is aligned with findings from previous study [5], which suggested that robustness evaluations involving adversarial versions of inappropriate prompts typically encompass two aspects: the model’s intrinsic generation robustness against the inappropriate prompt before any attack (pre-ASR) and its adversarial robustness against altered inappropriate prompts post-attack (commonly referred to as ASR, or post-ASR). Both pre-ASR and ASR metrics consistently reflect the robustness of unlearned DMs, and our results indicate that AdvUnlearn enhances both types of robustness evaluations, which contribute to aligned forgetting directions and demonstrate consistent unlearning performance of AdvUnlearn across both pre-ASR and ASR measures. **In the revision, we will make the above results and discussion clearer. Thanks for this insightful question.**

> [1] Circumventing Concept Erasure Methods For Text-to-Image Generative Model, ICLR 2023.

> [2] Hard Prompts Made Easy: Gradient-Based Discrete Optimization for Prompt Tuning and Discovery, NeurIPS 2023.

> [3] Prompting Hard or Hardly Prompting: Prompt Inversion for Text-to-Image Diffusion Model, CVPR 2024.

> [4] Ring-A-Bell! How Reliable are Concept Removal Methods For Diffusion Models?, ICLR 2024.

> [5] To Generate or Not? Safety-Driven Unlearned Diffusion Models Are Still Easy To Generate Unsafe Images ... For Now, ECCV 2024.

---

### Public Comment · ~Anh_Tuan_Bui2 · 2025-07-25
**Similar idea**

Dear authors,

Congrats on the great work! Interestingly, our work (also accepted to NeurIPS 2024) shares a similar adversarial training idea for concept erasure. So I’d like to take this chance to promote our work :D and help the community better understand the differences between the two approaches. :D We have included a comparison with AdvUnlearn in our arXiv paper, and would greatly appreciate it if the authors could consider updating your manuscript to reflect the relation between the two works.

**About our work**:

In our work, we propose an adversarial framework for concept erasure that formulates the problem as a bilevel min–max optimization. The core idea is to identify an adversarial concept that, when preserved, makes it more difficult for the model to retain the unwanted concept. We then fine-tune the UNet to erase the target concept while preserving the adversarial one—thereby improving both erasure quality and robustness. Rather than relying on a predefined retained set, our method adaptively selects what to preserve using a learned optimization strategy. We employ the Gumbel-Softmax trick to handle the discrete nature of concept selection, allowing us to optimize over token-based prompts in a differentiable way.

**Key differences from AdvUnlearn:**:

**Motivation**: AdvUnlearn is inspired by adversarial prompt attacks; our approach is driven by the empirical observation that the choice of preserved concept strongly affects erasure quality and model performance.

**Optimization Target**: AdvUnlearn fine-tunes the text encoder, while we fine-tune the UNet, directly influencing the image generation pipeline.

**Preservation Strategy**: AdvUnlearn requires a fixed retained set; our method adaptively selects the concept to preserve via Gumbel-Softmax.

**Optimization Approach**: We use Gumbel-Softmax, whereas AdvUnlearn uses FGSM for prompt search.

Link to the project page with code: https://tuananhbui89.github.io/projects/adversarial-preservation/

---

### Decision · Program_Chairs · 2024-09-25

**Decision:**

Accept (poster)

**Comment:**

**Summary:**
Diffusion models are highly effective in text-to-image generation but risk generating harmful content and violating copyrights. To address this, machine unlearning techniques, or concept erasing, have been developed but are vulnerable to adversarial prompt attacks. This work introduces AdvUnlearn, a robust unlearning framework using adversarial training (AT) to strengthen concept erasing. A direct use of AT can harm image quality, so a utility-retaining regularization is used to maintain balance. The text encoder, rather than UNet, is found more effective for robustification. AdvUnlearn shows superior robustness and maintains model utility, marking the first systematic exploration of robust DM unlearning with AT.

**Strengths:**
- Motivation:
  - "Significant" problem solving and "intuitive and reasonable" (gHPo)
  - "Relatively new and important" (4rR4)
- Results:
  - Relatively "thorough" experiment (gHPo)
  - "Enhanced robustness" maintaining "image quality" (hd8m)
  - "Plug-and-play usability" (hd8m)
- Clarity:
  - Formulated "clearly" (4rR4)

**Weaknesses:**
Initially, all the reviewers (gHPo, 4rR4, hd8m) shared concerns about insufficient model comparison, but the authors appear to have successfully addressed these issues. Reviewer 4rR4 raised a concern about the lack of quantitative results, which the authors thoroughly addressed in their rebuttal. As a result, Reviewer 4rR4 ultimately shifted to acceptance. The AC acknowledges the authors' persistent efforts and the active engagement of our reviewers.

**Justifications for Recommendation:**
Overall, the authors' rebuttal efforts were effective, and the major issues raised were well addressed. It appears that all major concerns from the reviewers were adequately resolved during the author-reviewer discussion period. Based on these, AC is pleased to recommend acceptance.